# Incorporating Consumer Ratings in Retailers' Discount Pricing of Digital Goods

Li Chen

Broadwell College of Business and Economics, Fayetteville State University, 1200 Murchison Road, Fayetteville, NC 28301, USA; lchen@uncfsu.edu

**Abstract**

Retailers of digital goods often use discount pricing to attract consumers. To make an effective promotion, they naturally want to understand consumers' valuation. Nonetheless, rigorous research is lacking on how to use consumer ratings on the retailer side. Our study aims to fill this research gap by investigating how retailers can determine optimal discount size in response to consumer ratings. We use both an analytical model and an empirical analysis. Our analytical results showed that discount size decreases with consumer ratings for non-supreme ratings. Nevertheless, there is no significant impact for supreme ratings. In addition, we find that consumers' confidence and the regular price of digital goods are critical moderators. Using a unique dataset of 419 online audiobooks, we empirically test the proposed hypotheses. The predictions of our model are consistent with empirical evidence. Our study demonstrates that retailers can provide smaller discounts when consumers give higher ratings of digital goods. In addition, consumers' confidence enlarges the consumer rating effect, while the regular price reduces such effect. Our findings can be applied to other digital goods such as digital movies, software/APP and online newspapers.

**Keywords:** consumer ratings; price discount; discount promotion; audiobooks; digital goods

## 1. Introduction

Digital goods have witnessed a rapid development in recent years [1]. Owing to the global adoption of the Internet and smart devices such as the iPhone and iPad, users can consume digital goods such as digital media, entertainment, and software applications quickly and conveniently [2]. With the market saturated with digital content vying for consumers' attention, retailers of digital goods have found it challenging to attract potential buyers [3].

Discount pricing is an efficient tool to promote physical products and services [4,5]. Not surprisingly, retailers of various digital goods (hereinafter "retailers") also apply this tool to attract potential consumers [6–8]). For example, in the digital movie market, Amazon Prime Video offered deals with discounted prices for renting (from $3.99) or buying (from $4.99) movies for 1000+ deals. For the software market, Norton antivirus software offered a deal of $19.99 for first year users (67% discount of the $59.99 regular price). In the online newspaper market, Wall Street Journal offered a Presidents Day sale, which charged $4/month for WSJ Digital for one year (90% discount of the $38.99 regular price) and $6/month for the WSJ Digital bundle for one year (88% discount of the $49.99 regular price). In the online streaming market, YouTube TV provided an offer of $49.99 for three months (40% discount of the $82.99 regular price) in August 2025.

From the retailers' point of view, there are two major differences between promoting digital goods and physical goods with a discount. First, consumers of digital goods usually only have one unit demand [9]. Therefore, they are more likely to be attracted by price signals than quantity-related tactics such as "Buy one get one free". Second, information technology developments enable digital goods to be distributed quickly and freely online [2,10]. As a result, retailers do not implement free delivery promotion [11] when promoting digital goods.

While discounts are widely used to facilitate digital goods sales, their effect will not be fully exploited if not properly designed [12,13]. Naturally, retailers desire to incorporate relevant, helpful, and quickly updated information to make better decisions such as the optimal discount size. We argue that one crucial, yet understudied factor is the prior consumer ratings [14]. Research reported that 90% of consumers read online ratings before making purchases [15]. While consumer ratings provide continuous and easily accessible information, prior studies mainly focus on the consumer side, such as how it influences consumers' value perception [16,17] and purchasing behavior [4,18].

Nevertheless, rigorous research is lacking on how to integrate consumer ratings on the retailer side. For example, little is known about when they will affect discount size and when they will not. In addition, prior research has identified key moderating factors of the consumer ratings effect, including reader's involvement and susceptibility [19] and products' quality signal [20]. Therefore, it is also critical to investigate potential moderating effects. Our study aims to fill this research gap. In particular, we seek to answer the following research questions:

*Research question 1: Will consumer ratings affect retailers' decision of discount size? If yes, what impacts will consumer ratings have on the discount size?*

*Research question 2: What factors will moderate the effect of consumer ratings on discount size?*

To address these research questions, we employ a two-period model [21] in which a retailer offers discount in the second time period. We derive the retailer's optimal discount size at equilibrium when prior consumer ratings are incorporated. Then, we conduct an empirical analysis based on real business data of an online audiobook retailer. We test our proposed hypotheses with multiple regression analysis and discuss the corresponding regression results.

Our main findings are as follows. First, our analytical model shows that whether consumer ratings affect the retailer's discount size depends on rating values. To be more precise, discount size is negatively associated with non-supreme consumer ratings but not with supreme ratings. Second, we find that consumers' confidence (using the number of raters as a proxy) and the regular price of digital goods will moderate the effect of consumer ratings. In particular, while consumers' confidence increases the negative impact of consumer ratings, the regular price reduces such impact. Third, our empirical analysis results based on real online business data of audiobooks are consistent with our proposed predictions. Overall, our paper makes an exploratory study on how retailers can incorporate consumer ratings in their discount size decisions.

Our study contributes to the e-commerce literature in the following aspects. First, we develop an analytical model to build theoretical backgrounds for retailers to use consumer ratings when determining discount size. Our model complements prior research by providing new insights for retailers to incorporate consumers' feedback. Second, this study adds to the existing studies of online consumer ratings because our results demonstrate that the rating effect varies across different rating values. We also provide a framework for retailers to inspect key moderating factors. Third, we test our proposed hypotheses with real business data in the online audiobooks market. This approach provides a comprehensive investigation on this topic.

The rest of the paper is organized as follows. We provide a comprehensive review of relevant research in Section 2. Section 3 presents the analytical model under which the retailer determines the optimal discount size after incorporating consumer ratings. Then, we derive a set of hypotheses based on our analytical results. Using a unique dataset of 419 audiobooks collected from an online audiobook retailer, we conduct an empirical analysis to test the proposed hypotheses in Section 4. We then present the regression analysis results and the robustness check analysis results. Section 5 discusses the theoretical and managerial implications of our study. We conclude this paper with limitations and future research directions in Section 6.

## 2. Literature Review

In this section, we discuss two streams of the relevant literature, online consumer ratings and discount pricing.

### 2.1. Online Consumer Ratings

Our paper follows the literature that investigates online consumer ratings. Prior research has shown that online consumer ratings can help consumers to evaluate the quality of products and services [22,23], build consumers' trust [24,25], and reduce consumers' uncertainty, especially on experience goods [16,18,26]. As a result, consumers will check ratings and reviews left by prior consumers before making a purchasing decision online [27,28].

Not surprisingly, a large number of researchers focus on the impact of online consumer ratings on sales of digital goods [29]. Ref. [30] found that firms with high online ratings have advantages in market competitions. Ref. [31] reported that consumers respond differently to prior reviews of video games. In particular, two features of videogames showed significant moderating effects: the violence level and social orientation. Ref. [32] found that the volume of consumer ratings related to product quality and ease-of-use drives more sales than the valence of consumer ratings. Ref. [33] showed that the helpfulness of consumer reviews played a critical role in the sales of newly released movies.

Nevertheless, researchers have pointed out that consumer ratings might not always be beneficial for retailers. For example, previous studies have reported that negative consumer reviews and ratings had a greater effect than positive consumer reviews and ratings, reflecting a negativity bias [34,35]. Ref. [17] found that the low credibility of retailers' electronic word-of-mouth can reduce consumers' purchase intentions.

While this stream of research has provided general knowledge of online consumer ratings, the dominant majority of the extant research deals with how consumers respond to online ratings. Little research attention has been paid to its potential use on the retailer side of the market. Unlike these studies, our research explores how retailers can make use of online consumer ratings in their decisions on discount size.

### 2.2. Discount Pricing

Our study is also aligned with the literature of discount pricing. In this stream of research, one group of studies concentrated on developing optimal policies of offering discounts. For example, Ref. [36] examined the optimal price discount mechanisms for consignment contracts in a dynamic supply chain. Ref. [37] analyzed when retailers should provide their own discount promotion or participate in e-commerce platforms' promotion events. Ref. [38] proposed an optimal revenue-management model incorporating add-on discounts and online learning algorithms. Their numerical experiments based on real transaction data of video games showed robust performance improvement. Ref. [39] investigated retailers' optimal discount policy in the online streaming market under a resale model and an agency model.

Another group of studies examined the impact of discounts on retailers' sales. Within this line of research, Ref. [10] found that price promotion increases digital movie sales as well as digital rental sales. In the same vein, Ref. [6] showed that both discount rate and discounted price have positive relationship with sales increase in the game-as-a-service (GAAS) application market. Based on their experimental results, Ref. [7] reported that retailers can update consumers' needs with deep discounts to enhance their purchase intentions. Observing the behavior of real online game players, Ref. [8] found that price promotions are, in general, profitable for online game providers. Ref. [40] showed that prior discount information plays an important role in motivating B2B software salespersons.

While discount pricing has been widely considered as a driver of sales incrementation in the market, it might have negative effects as well. Ref. [12] reported that inauthentic discounts will hurt the symbolic value of the products. Ref. [13] found that price promotions such as discount pricing may arouse quality concerns of consumers and weaken the effectiveness of retailers' referral marketing. Ref. [41] reported that large discount sizes from retailers might lead to a higher level of uncertainty on perceived quality among consumers.

Unlike the above studies, we first develop a model to incorporate consumer ratings in retailers' discount decisions and derive a set of relevant hypotheses. Then, we test these hypotheses with real business data. So far, little research in this area has incorporated both analytical and empirical analysis. Ref. [42] proposed a dynamic pricing model in response to consumer ratings and reviews. However, their study did not include an empirical analysis. In addition, retailers in their model introduced a higher price in the later stage, which is inconsistent with our observation of retailers' promotion practices. Table 1 below summarizes the difference between our study and recent relevant literature.

**Table 1.** Summary of recent relevant literature.

| Authors | Research Context | Analytical Model | Discount Size | Consumers Ratings | Moderating Factors | Empirical Analysis |
|---|---|---|---|---|---|---|
| Gong et al. (2015) [10] | Digital movie | | ✓ | | | ✓ |
| He and Chen (2018) [42] | Electronic platform | ✓ | | ✓ | | |
| Li et al. (2020) [21] | Online coupon strategies | ✓ | ✓ | | ✓ | |
| Runge et al. (2022) [8] | Online videogame | | ✓ | ✓ | | ✓ |
| Bergers et al. (2023) [40] | Software | | ✓ | | ✓ | ✓ |
| Zhang et al. (2024) [26] | E-commerce websites | | ✓ | ✓ | | ✓ |
| Our study | Online audiobook | ✓ | ✓ | ✓ | ✓ | ✓ |

## 3. Analytical Model

### 3.1. Model Setup

In this section, we construct an analytical model that examines how retailers set their optimal discount size after incorporating consumers' online ratings. We set up a two-period framework in which retailers offer discounts in the second time period. This framework is widely used in the prior literature [21,43,44]. We made several common assumptions as follows:

**Assumption 1.** *Consumers are uniformly distributed between [0, 1].*

**Assumption 2.** *Digital goods have negligible marginal production cost.*

**Assumption 3.** *Consumers have one unit demand of digital goods and there is no resale channel of digital goods.*

We do not adopt the model in which retailers set a lower initial price and increase it later. First, internet-savvy consumers usually search for all deals available before making purchases [45]. Since these consumers experience a high searching cost, they are eager to pay

a lower price. Second, it is consistent with most of our observations that retailers set regular prices first and provide a discount later. Figure 1 below presents our model framework.

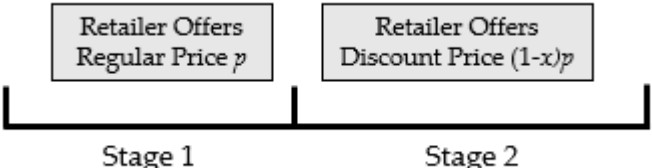

**Figure 1.** Model framework.

While retailers only determine the optimal discount size in stage 2, stage 1 is also a critical factor in our model. First, consumers set initial expected quality in stage 1, which serves as the basis of their updated quality evaluation. They also need a stage to post their ratings. Second, retailers need a regular price stage to collect prior consumer ratings so they can incorporate this information in their discount decisions.

Following the literature, we assume that there are two types of consumers, myopic consumers and strategic consumers [21]. While myopic consumers primarily rely on their own expectations in the first time period, strategic consumers make their decisions after reading prior consumer ratings and consider the discount offered by the retailer in the second time period.

In the first time period, let $q_e \in [0, 1]$ denote the myopic consumers' expected intrinsic value of the digital good and $p$ denote the regular price of the digital good. Please note that myopic consumers depend on their intrinsic value because there are no prior consumer ratings at that time. To incorporate consumers' preferences on product attributes, we use a taste parameter $s \in [0, 1]$ to present the attribute position. Consumers' utility will decrease by $k$ per unit distance, which is similar to the travel distance in the Hotelling model [46]. Therefore, consumers' utility is

$$U = q_e - p - ks$$

After those early adapters consume the digital good, they post their ratings. Thus, in the second time period, strategic consumers have an opportunity to update their value expectation based on the peer opinions. For example, when the prior ratings are higher than the initial expectation, the updated value perception will go upward. On the contrary, the updated value perception will decrease when prior ratings signal lower evaluation than expected.

Let $r$ denote the valence of prior consumer ratings, $\beta$ denote the level of consumers' confidence towards prior ratings [47], and $\lambda$ denote consumers' sensitivity towards the change in value expectation. Then, the consumers' updated value perception $q_u$ is

$$q_u = q_e + \lambda(\beta r - q_e)$$

In addition, we denote $x$ as the discount size. As a result, $(1 - x)p$ denotes the discount price offered by the retailer. For example, if the regular price is $19.99, and the price after discount is $7.99, then $x = 60\%$. Let $c$ refer to the waiting cost. Correspondingly, consumers' utility becomes

$$U = q_u - (1 - x)p - ks - c$$

Table 2 below summarizes the notations used in the model.

**Table 2.** Summary of notations.

| Notation | Definition |
| --- | --- |
| $x$ | Discount size |
| $q$ | Intrinsic value provided by the digital good |

**Table 2.** *Cont.*

| Notation | Definition |
|----------|-----------|
| $p$ | Regular price |
| $k$ | Unfit cost of distance |
| $r$ | Value of consumer ratings |
| $s$ | Feature position |
| $\lambda$ | Sensitivity towards the change in consumers' value expectation |
| $\beta$ | Consumers' confidence in prior ratings |
| $m$ | Ratio of consumers in the second period over that in the first period |

*3.2. Equilibrium Results*

We first identify the upper and lower threshold of prior consumer ratings $r$, in which the consumers' updated value perception is between [0, 1]. The rationale is to control the impact of extreme values of consumer ratings [22]. Lemma 1 below describes the identified thresholds.

**Lemma 1** (*Results for extreme ratings*). *When the prior consumer ratings value exceeds the upper threshold $r \geq \frac{1-(1-\lambda)q_e}{\lambda\beta}$, consumers' updated value perception reaches the maximum ($q_u = 1$). The retailer's optimal discount size and total revenue are as follows: $x^* = 1 - \frac{1-c}{2p}$ and $\pi_t = \frac{p(q_e-p)}{k} + \frac{m(1-x)\times p\times(1-(1-x)\times p-c)}{k}$. When the prior consumer ratings value is smaller than the lower threshold $r \leq \frac{(\lambda-1)q_e}{\lambda\beta}$, their value perception becomes zero ($q_u = 0$). Thus, there is no demand. (Please see the proof in Appendix A).*

Then, we investigate the case in which the prior consumer ratings value is between the thresholds in Lemma 1 ($\frac{(\lambda-1)q_e}{\lambda\beta} < r < \frac{1-(1-\lambda)q_e}{\lambda\beta}$). In this case, consumers' updated value perception is between [0, 1] ($0 < q_u < 1$). As a result, consumers' utility becomes $U = (q_e + \lambda(\beta r - q_e))(1-x) \times p - c - ks$. Using the similar approach of Lemma 1, we obtain the equilibrium results and obtain the following Lemma:

**Lemma 2** (*Results for non-extreme ratings*). *When prior consumer ratings lead to updated value perception between [0, 1], the retailer's optimal discount size and total revenue are as follows: $x^* = \frac{c+2p-q_e+\lambda q_e-\lambda\beta r}{2p}$, and $\pi_t = \frac{p(-p+q_e)}{k} + \frac{m(1-x)p(-c+q_e+\lambda(-q_e+\beta r)-p(1-x))}{k}$. (Please see the proof in Appendix A).*

Using the results of Lemmas 1 and 2 above, we are able to analyze the impact of consumer ratings on the retailer's discount size at equilibrium. A de tailed inspection yields the following proposition:

**Proposition 1** (*Effect of consumer ratings on discount size*). *When prior consumer ratings $r$ is between the lower threshold and the upper threshold mentioned in Lemma 1, retailer's optimal discount size $x^*$ decreases with the prior ratings. Nevertheless, when prior consumer ratings $r$ goes beyond those two thresholds, retailer's optimal discount size $x^*$ will not be affected by the prior ratings. (Please see the proof in Appendix A).*

Proposition 1 reflects how prior consumer ratings affect retailers' optimal discount. Based on Lemma 2 results, $\frac{\partial x}{\partial r} = -\frac{\lambda\beta}{2p} < 0$ when taking the derivative of optimal discount size $x^*$ over consumer ratings $r$. However, $\frac{\partial x}{\partial r} = 0$ under Lemma 1.

Then, we investigate the moderating effects of two factors, (1) consumers' confidence, and (2) the regular price of the digital goods. We use the number of raters as a proxy of consumers' confidence. Ref. [24] showed that a large number of reviews in the positive

summary plays a moderating role on consumers' trust of E-tailer websites. Ref. [48] reported that the online rating score on the tourism website TripAdvisor is related to the number of reviewers. Proposition 2 below describes the moderating effects.

**Proposition 2** (*Moderating effects*). *Given a fixed value of consumers' sensitivity* $\lambda$, *consumers' confidence enlarges the negative impact of consumer ratings. On the other hand, the regular price of the digital good reduces the negative impact of consumer ratings. (Please see the proof in Appendix A).*

*3.3. Hypotheses*

Then, we derived a set of hypotheses from our findings above, which can be tested with an empirical study. We collected data from an online retailer of audiobooks. The process of data collection will be described in detail in the next section. Figure 2 below presents the conceptual model of our hypotheses. Please note that Figure 2 is based on our analytical results, not on Figure 1. In addition, H2 and H3 are based on digital goods with non-supreme ratings. For the digital goods with supreme ratings, our results reflect that there is no significant association between consumer ratings and discount size.

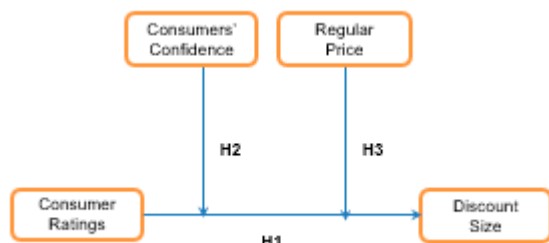

**Figure 2.** Conceptual model of hypotheses.

Based on Proposition 1, we propose the following hypotheses related to the effect of consumer ratings on retailer's discount size decision:

**Hypothesis 1A.** *Discount sizes are negatively associated with the consumer ratings of digital goods when they have non-supreme ratings (ratings within the thresholds of Lemma 1).*

**Hypothesis 1B.** *Discount sizes are not affected by the consumer ratings of digital goods when they have supreme ratings (ratings beyond the thresholds of Lemma 1).*

Based on Proposition 2, we propose the following hypotheses related to moderating factors. We argue that the number of raters serves as an ideal proxy of confidence in the context of audiobook market. While consumers might be uncertain of an average rating with only two or three raters, an average rating from a large number of raters, such as 2000 or 3000 raters, can be more persuasive. The more prior buyers involved in a rating, the higher level of confidence consumers will have.

**Hypothesis 2.** *The effect of consumer ratings on discount size of digital goods is smaller when there is a larger number of raters.*

As for the factor of regular price, our analytical results reflect that it exerts a reducing moderating effect for audiobooks. Therefore, we propose the following hypothesis:

**Hypothesis 3.** *The effect of consumer ratings on discount size of digital goods is larger for those with a higher regular price.*

## 4. Empirical Analysis

Our analytical model in the last section explores retailers' discount decisions in response to consumer ratings. In this section, we empirically test our hypotheses using data of audiobooks. The audiobook market was chosen as our research context for the following reasons. First, audiobooks are one of the most rapidly growing markets of digital goods. Ref. [49] reported that the sales of audiobooks have witnessed 11 consecutive years of double-digit growth in 2023. Audiobook sales have been projected to reach $15 billion in 2027 [50]. Nonetheless, retailers of some other digital goods, such as online streaming retailers Disney+ and Hulu, have seen declines in subscriber growth [51]. Second, audiobooks belong to experience goods of long-time consumption. It usually takes readers several days or even weeks to finish a book. Thus, potential buyers want to read prior consumer ratings before buying the books. For digital goods like online music, consumers will just spend several minutes listening to it. Third, prices of audiobooks are usually much higher than print books and e-books because publishers often invite popular narrators and incorporate specially created soundscapes [52]. It is reasonable for consumers to seek deals in the market, which justifies retailers' discount strategy. Based on the discussion above, we believe that the audiobook market provides an appropriate setting to test our hypotheses.

### 4.1. Data

To test our hypotheses, we need to observe retailers offering price discounts as well as consumer ratings. We chose an online audiobook retailer Chirpbooks, which offers discounts from time to time in order to have its audiobooks discovered by consumers. Unlike Audible and Audiobooks, Chirpbooks is a non-subscription audiobook platform with intermittent price promotion. It does not require a monthly subscription fee, which enables us to focus on the effect of consumer ratings on discount sizes without worrying about the impact of a subscription model on consumers' purchases.

Please note that Chirpbooks is different from deal-of-the-day platforms and algorithmic pricing with reputation signals. For deal-of-the-day platforms, the majority of the deals are with discounts. However, Chirpbooks provides audiobooks both with discounts and without discounts. For the algorithmic pricing with reputation signals, this model can lead to prices both higher than or lower than the regular price. However, Chirpbooks only provides discounted prices, not prices higher than the regular price.

For each audiobook with a discount on Chirpbooks, we manually collected the following information: the regular price and the price after the discount, consumer ratings, the number of raters, and related book features. Our dataset includes 419 audiobooks from January 2022 to February 2022. Prior research uses datasets of a similar size. For example, Ref. [53] collected online consumer ratings of 261 high-end hotels. Ref. [54] collected the ratings of critics and consumers on 408 digital movie titles.

*Dependent variable*

The dependent variable is the discount size, which is measured as the percentage of the regular price of an audiobook that has been discounted. For example, if the regular price of an audiobook is $19.99 and the price after discount is $11.99, then the discount size is 40%.

*Independent variables*

The key independent variables include consumer ratings, the number of raters, and the regular price. We use the log transformation of the number of raters and the regular price to control the effect of overdispersion.

*Control variables*

Following the literature, we use a set of control variables to improve estimation precision. At the book level, we include the length of audiobook (log of the length minutes)

and the age of a book (log of the number of years after release). In addition, we use a dummy variable to control whether the book is on the *New York Times* bestseller list [55]. We use another dummy variable to control whether an audiobook is published by one of the big five publishers (Penguin, Hachette Book Group, Harper Collins, Simon and Schuster, and Macmillan). Table 3 below presents the summary statistics of our data.

**Table 3.** Summary statistics.

| Variable | Definition | Min | Mean | Max | Standard Deviation |
|---|---|---|---|---|---|
| Discount_size | The percentage of the regular price discounted | 0.572 | 0.835 | 0.960 | 0.062 |
| Rating | Average consumer ratings | 3.300 | 4.420 | 5.000 | 0.275 |
| Ln_Raters | Log of the number of raters | 2.303 | 3.369 | 6.686 | 0.894 |
| Ln_Regprice | Log of the regular price of a book | 1.384 | 2.954 | 4.190 | 0.350 |
| Ln_Length | Log of the number of minutes of a book | 4.691 | 6.324 | 7.741 | 0.423 |
| Ln_BookAge | Log of number of years since publication | 0 | 1.625 | 3.332 | 0.722 |
| Bestseller_Dummy | 1 for bestseller, 0 otherwise | 0 | 0.317 | 1 | 0.466 |
| BigPub_Dummy | 1 for big five publishers, 0 otherwise | 0 | 0.224 | 1 | 0.418 |

We have checked the correlation values among variables used in our model. Table 4 below presents the correlation matrix. We find that most correlation values are small (<0.33), which implies that there is no extremely close relationship.

**Table 4.** Correlation matrix of variables.

| | 1 | 2 | 3 | 4 | 5 | 6 | 7 | 8 |
|---|---|---|---|---|---|---|---|---|
| 1. Discount_size | 1 | | | | | | | |
| 2. Rating | −0.27 | 1 | | | | | | |
| 3. Ln_Raters | 0.10 | 0.18 | 1 | | | | | |
| 4. Ln_Regprice | 0.31 | 0.00 | −0.11 | 1 | | | | |
| 5. Bestseller_Dummy | −0.21 | 0.13 | −0.02 | 0.08 | 1 | | | |
| 6. Ln_Length | 0.03 | 0.11 | 0.02 | 0.32 | 0.10 | 1 | | |
| 7. Ln_BookAge | −0.25 | 0.19 | 0.00 | 0.11 | 0.30 | 0.06 | 1 | |
| 8. BigPub_Dummy | −0.25 | 0.06 | −0.05 | 0.30 | 0.16 | 0.06 | 0.16 | 1 |

### 4.2. Empirical Test on the Impact of Consumer Ratings

We first check the impact of consumer ratings. To test Hypothesis 1 empirically, we develop the following model (Model 1):

$$Discount = \beta_0 + \beta_1\, Rating + \beta_2\, Ln\_Raters + \beta_3\, Ln\_Regprice + \beta_4\, Bestseller\_Dummy$$
$$+ \beta_5\, Ln\_Length + \beta_6\, Ln\_BookAge + \beta_7\, BigPub\_Dummy + \varepsilon$$

The key independent variable is consumer ratings. Nevertheless, Hypothesis 1 suggests that consumer ratings will have a significant impact for audiobooks with non-supreme ratings (H1A), but not for audiobooks with supreme ratings (H1B). Thus, a regression analysis with all audiobooks will not deliver valid testing results. To test H1A and H1B, we divide our dataset into two groups. Group one includes 388 audiobooks with consumer ratings below 4.8 to test H1A and group two includes 31 audiobooks with consumer ratings 4.8 and higher to test H1B.

We chose 4.8 as the threshold of supreme rating because audiobooks with a rating of 4.8 and higher are usually considered to be of superior quality. Ref. [20] reported that extreme negative ratings comprise 9.47% of all ratings. This is similar to the percentage of supreme ratings in our study. We do not study extremely low ratings because the minimum rating in our dataset is 3.3.

H1A suggests that discount size is negatively associated with consumer ratings. Therefore, we predict that its coefficient ($\beta_1$) will be negative when applying our model. To

provide a comprehensive and robust analysis, we present three models with different control variables. Table 5 below presents the regression results. Model 1-1 of column (1) only includes the book age and book length, and Model 1-2 of column (2) only includes the best seller dummy and the big publisher dummy. Model 1-3 of column (3) includes all control variables.

**Table 5.** Effect of consumer ratings (medium-rating audiobooks N = 388).

|  | (1) | (2) | (3) |
|---|---|---|---|
| *Rating* | −0.045 (0.011) *** | −0.054 (0.010) *** | −0.045 (0.010) *** |
| *Ln_Raters* | 0.012 (0.003) *** | 0.011 (0.003) *** | 0.011 (0.003) *** |
| *Ln_Regprice* | 0.072 (0.008) *** | 0.083 (0.008) *** | 0.088 (0.008) *** |
| *Ln_Length* | −0.007 (0.007) |  | −0.010 (0.007) |
| *Ln_BookAge* | −0.023 (0.004) *** |  | −0.016 (0.004) *** |
| *Bestseller_Dummy* |  | −0.019 (0.006) *** | −0.011 (0.006) * |
| *BigPub_Dummy* |  | −0.049 (0.007) *** | −0.047 (0.007) *** |
| *(Intercept)* | 0.859 (0.063) *** | 0.798 (0.050) *** | 0.835 (0.059) *** |
| *Adjusted R-square* | 0.264 | 0.333 | 0.360 |

Notes: Standard errors are reported in parentheses. *** $p < 0.01$ * $p < 0.1$.

Then, we apply the same models on audiobooks with supreme ratings. Based on H1B, the coefficient $\beta 1$ will not show a significant impact. Table 6 below presents the regression results.

**Table 6.** Effect of consumer ratings (premium-rating audiobooks N = 31).

|  | (1) | (2) | (3) |
|---|---|---|---|
| *Rating* | 0.133 (0.120) | −0.030 (0.102) | 0.003 (0.008) |
| *Ln_Raters* | 0.014 (0.012) | −0.010 (0.010) | −0.000 (0.001) |
| *Ln_Regprice* | 0.135 (0.042) *** | 0.119 (0.035) *** | 0.009 (0.003) ** |
| *Ln_Length* | 0.039 (0.026) |  | 0.004 (0.002) ** |
| *Ln_BookAge* | −0.031 (0.012) ** |  | −0.003 (0.001) *** |
| *Bestseller_Dummy* |  | −0.006 (0.013) | −0.001 (0.001) |
| *BigPub_Dummy* |  | −0.065 (0.016) *** | −0.006 (0.001) *** |
| *(Intercept)* | −0.517 (0.620) | 0.630 (0.538) | −0.018 (0.043) |
| *Adjusted R-square* | 0.554 | 0.674 | 0.812 |

Notes: Standard errors are reported in parentheses. *** $p < 0.01$, ** $p < 0.05$,.

Regarding the coefficient of consumer ratings (the first row in Table 5), we can see that it is negative and significant ($p < 0.05$ in all three models) for audiobooks with non- supreme ratings. Therefore, H1A is supported. According to the estimates of column (3), for two audiobooks with the same features, an audiobook with one unit higher in consumer ratings has a 4.7% lower discount size. This effect is substantial because the standard deviation for the discount size is only 0.062. In addition, the coefficient of consumer ratings (the first row in Table 6) reflects an insignificant impact for audiobooks with premium ratings ($p > 0.1$ in all three models). Therefore, H1B is supported. One recommendation for retailers is to consider different discount policies for audiobooks with different rating levels.

In addition, we notice that the book age and whether a book is published by a big five publisher exert a significant impact in both Tables 5 and 6. In particular, retailers would like to offer a smaller discount for old books and books published by big five publishers.

### 4.3. Empirical Test on Moderating Effects

Next, we empirically test the moderating effects by introducing two dummy variables: *HRaters_Dummy* and *HPrice_Dummy*. Using dummy variables to explore moderating roles is frequently carried out in the literature [22,31,56]. The dummy variable *HRaters_Dummy* equals one when the number of raters is higher than the average rater number (rounded to 52), and zero otherwise. Similarly, the dummy variable *HPrice_Dummy* equals one when the regular price is higher than the average regular price (rounded to \$20), and zero otherwise. Please note that these two moderating factors are independent. First, our analytical results do not reflect that they are interrelated. Second, no prior research has reported dependency between these two moderators. We develop the following model (Model 2) to test Hypotheses 2 and 3:

$$
\begin{aligned}
Discount = {} & \beta 0 + \beta 1\ Rating + \beta 2\ HRaters\_Dummy \times Rating + \beta 3\ HRaters\_Dummy \\
& + \beta 4\ HPrice\_Dummy \times Rating + \beta 5\ HPrice\_Dummy + \beta 6\ Ln\_Length \\
& + \beta 7\ Ln\_BookAge + \beta 8\ Bestseller\_Dummy + \beta 9\ BigPub\_Dummy + \varepsilon
\end{aligned}
$$

Hypothesis 2 suggests that the number of raters enlarges the negative impact of consumer ratings on discount size. Therefore, we predict that the coefficient $\beta 2$ will be negative. Similarly, Hypothesis 3 reflects that the regular price reduces the impact of consumer ratings. Therefore, we predict that the coefficient $\beta 4$ will be positive. Table 7 below presents the regression results. To provide a comprehensive analysis, we use three models. Model 2-1 of column (1) only includes *HRaters_Dummy* and its interaction with consumer ratings, and Model 2-2 of column (2) only includes *HPrice_Dummy* and its interaction with consumer ratings. Model 2-3 of column (3) includes all variables. Please note that we only use 388 audiobooks with non-supreme ratings.

Column (1) of Table 7 shows the estimates where we only include the moderating effect of consumers' confidence. We can see that the coefficient estimate $\beta 2$ is negative and statistically significant ($p < 0.01$), which is consistent with our prediction. Similar results are reflected in column (3) as well. Therefore, H2 is supported. Consequently, retailers can take advantage of larger number of raters by providing smaller discount sizes.

Column (2) of Table 7 represents the estimates where we only include the moderating effect of the regular price. The coefficient estimate $\beta 4$ is positive and statistically significant ($p < 0.01$), which is consistent with our prediction. Similar results are reflected in column (3) as well. Therefore, H3 is supported. In practice, retailers can offer larger discount sizes for audiobooks with higher regular prices.

Column (3) of Table 7 shows the estimates where we include the moderating effects of both consumers' confidence and the regular price. The corresponding coefficients show that both factors moderate the impact of consumer ratings. According to the estimates

in Column (3), given that all other factors are fixed, when the number of raters changes from low to high, the discount size will decrease by 6.5%. Similarly, given that all other factors are fixed, when the regular price changes from low to high, the discount size will increase by 7.3%. Overall, our results have shown consistent and significant support for the proposed hypotheses.

**Table 7.** Moderating effects of consumers' confidence and regular price (N = 388).

|  | **(1)** | **(2)** | **(3)** |
|---|---|---|---|
| *Rating* | −0.027 (0.013) ** | −0.079 (0.015) *** | −0.068 (0.015) *** |
| *HRaters_Dummy * Rating* | −0.077 (0.028) *** |  | −0.065 (0.025) ** |
| *HRaters_Dummy* | 0.349 (0.122) *** |  | 0.304 (0.112) *** |
| *HPrice_Dummy * Rating* |  | 0.072 (0.022) *** | 0.073 (0.021) *** |
| *HPrice_Dummy* |  | −0.275 (0.095) *** | −0.280 (0.094) *** |
| *Ln_Length* | 0.011 (0.007) | 0.003 (0.007) | 0.003 (0.007) |
| *Ln_BookAge* | −0.012 (0.004) *** | −0.010 (0.004) ** | −0.008 (0.004) ** |
| *Bestseller_Dummy* | −0.014 (0.007) ** | −0.014 (0.006) ** | −0.012 (0.006) ** |
| *BigPub_Dummy* | −0.030 (0.007) *** | −0.042 (0.007) *** | −0.042 (0.007) *** |
| *(Intercept)* | 0.915 (0.068) *** | 1.172 (0.070) *** | 1.118 (0.073) *** |
| *Adjusted R-square* | 0.162 | 0.268 | 0.289 |

Notes: Standard errors are reported in parentheses. *** $p < 0.01$ ** $p < 0.05$ * $p < 0.1$.

While we have inspected audiobooks, our findings can be easily applied to other digital goods such as e-books, online newspapers, digital movies, etc. First, consumer ratings of digital goods are widely available online. Retailers can easily find them either on their own platform or relevant social media and online forums. Second, our analytical results are based on common assumptions. The corresponding empirical analysis uses variables that are simple to observe in online platforms, making it convenient for retailers to conduct their own analysis.

### 4.4. Robustness Check

To validate the robustness of our results, we performed three robustness check analyses of our findings. First, it is vital to make sure that our results are robust to different measures of the threshold of supreme rating. Therefore, we rerun our regression analysis using 4.7 as the threshold to separate our dataset. In this case, we have 336 audiobooks with consumer ratings lower than 4.7 (non-supreme rating), and 83 audiobooks with consumer ratings 4.7 and higher (supreme rating). We do not use 4.9 as the threshold because we will have very few data points for supreme-rating audiobooks.

The results of this robustness check analysis are reported in Tables 8 and 9. Our major findings hold qualitatively. The Table 8 results show that consumer ratings significantly

and negatively affect the discount size of books with non-supreme ratings. Nonetheless, such an effect is absent for audiobooks with supreme ratings (please see Table 9 below).

**Table 8.** Robustness check results (Non-supreme rating Audiobooks N = 336).

|  | (1) | (2) | (3) |
|---|---|---|---|
| *Rating* | −0.033 (0.012) *** | −0.035 (0.012) *** | −0.029 (0.012) ** |
| *Ln_Raters* | 0.013 (0.003) *** | 0.011 (0.003) *** | 0.012 (0.003) *** |
| *Ln_Regprice* | 0.059 (0.009) *** | 0.071 (0.011) *** | 0.076 (0.009) *** |
| *Ln_Length* | −0.005 (0.008) | | −0.008 (0.007) |
| *Ln_BookAge* | −0.018 (0.005) *** | | −0.012 (0.004) *** |
| *Bestseller_Dummy* | | −0.015 (0.006) ** | −0.009 (0.007) |
| *BigPub_Dummy* | | −0.050 (0.007) *** | −0.048 (0.007) *** |
| *(Intercept)* | 0.817 (0.069) *** | 0.750 (0.055) *** | 0.784 (0.065) *** |
| *Adjusted R-square* | 0.176 | 0.272 | 0.288 |

Notes: Standard errors are reported in parentheses. *** $p < 0.01$ ** $p < 0.05$.

**Table 9.** Robustness check results (supreme-rating audiobooks N = 83).

|  | (1) | (2) | (3) |
|---|---|---|---|
| *Rating* | 0.021 (0.065) | −0.013 (0.063) | 0.025 (0.057) |
| *Ln_Raters* | 0.008 (0.006) | 0.006 (0.006) | 0.006 (0.005) |
| *Ln_Regprice* | 0.131 (0.018) *** | 0.148 (0.017) *** | 0.140 (0.016) *** |
| *Ln_Length* | −0.007 (0.014) | | −0.010 (0.012) |
| *Ln_BookAge* | −0.037 (0.007) *** | | −0.029 (0.007) *** |
| *Bestseller_Dummy* | | −0.028 (0.010) *** | −0.013 (0.009) |
| *BigPub_Dummy* | | −0.053 (0.012) *** | −0.049 (0.011) *** |
| *(Intercept)* | −0.409 (0.328) | 0.425 (0.314) | 0.394 (0.287) |
| *Adjusted R-square* | 0.581 | 0.603 | 0.681 |

Notes: Standard errors are reported in parentheses. *** $p < 0.01$.

Second, consumers are often concerned about the reliability of online ratings [57]. As a result, they might be cautious when viewing prior ratings with high confidence (using the number of raters as proxy). We thus check the measure of the 80th percentile instead of the average of the number of raters and the regular price. This is because the average number

of consumer raters (52) used in the last subsection is already close to the third quartile (75th percentile) in our dataset. Therefore, we believe that the 80th percentile reflects a more conservative measure.

We redefine the two dummy variables used in model 2 to reflect moderating effects. Thus, *HRaters_Dummy* equals one when the number of raters is higher than the 80th percentile of the rater number in our dataset, which is rounded to 62, and zero otherwise. Similarly, *HPrice_Dummy* equals one when the regular price of an audiobook is higher than the 80th percentile of the regular price in our dataset, which is rounded to $23, and zero otherwise. Table 10 below presents the results of the robustness check. We can see that both the rater number and the regular price reflect an expected significant moderating effect.

**Table 10.** Robustness check results of moderating effects (N = 336).

|  | (1) | (2) | (3) |
|---|---|---|---|
| *Rating* | −0.030 (0.013) ** | −0.066 (0.014) *** | −0.055 (0.014) *** |
| *HRaters_Dummy \* Rating* | −0.078 (0.031) ** |  | −0.061 (0.038) ** |
| *HRaters_Dummy* | 0.359 (0.138) *** |  | 0.290 (0.129) ** |
| *HPrice_Dummy \* Rating* |  | 0.061 (0.023) *** | 0.056 (0.023) ** |
| *HPrice_Dummy* |  | −0.227 (0.102) ** | −0.203 (0.101) ** |
| *Ln_Length* | 0.013 (0.007) * | 0.003 (0.007) | 0.003 (0.007) |
| *Ln_BookAge* | −0.015 (0.005) *** | −0.012 (0.004) *** | −0.012 (0.004) *** |
| *Bestseller_Dummy* | −0.010 (0.007) | −0.012 (0.006) * | −0.010 (0.006) |
| *BigPub_Dummy* | −0.027 (0.007) *** | −0.037 (0.007) *** | −0.036 (0.007) *** |
| *(Intercept)* | 0.905 (0.065) *** | 1.111 (0.070) *** | 1.065 (0.072) *** |
| *Adjusted R-square* | 0.165 | 0.254 | 0.279 |

Notes: Standard errors are reported in parentheses. *** $p < 0.01$ ** $p < 0.05$ * $p < 0.1$.

Furthermore, we use the two-stage-least-square (2SLS) method to test the potential simultaneity issue. While the consumer ratings influence discount size, the discounted price might influence consumer ratings. We use a dummy variable AfterCOVID_Dummy as an instrumental variable (IV) which equals one if a book was published in 2020 or after 2020 and zero otherwise. The rationale here is that the eruption of COVID-19 was an exogenous event/shock. After the pandemic, people have experienced a significant change, affecting their evaluation of digital goods, including audiobooks. Nonetheless, it is unlikely to influence retailers' discount decisions. The results of our analysis are shown in Table 11 below. We can see that the significance and the sign of the consumer ratings effect remains the same.

**Table 11.** 2SLS Analysis results (non-supreme-rating audiobooks N = 388).

|  | 2SLS Regression |
| --- | --- |
| *Rating* | −0.614 (0.303) ** |
| *HRaters_Dummy \* Rating* | −0.011 (0.045) |
| *Ln_Raters* | 0.060 (0.195) |
| *HPrice_Dummy \* Rating* | 0.186 (0.136) |
| *Ln_Regprice* | −0.724 (0.595) |
| *Ln_Length* | −0.002 (0.008) |
| *Ln_BookAge* | −0.009 (0.006) |
| *Bestseller_Dummy* | −0.014 (0.006) |
| *BigPub_Dummy* | −0.050 (0.007) *** |
| *(Intercept)* | 3.267 (1.300) ** |
| *Adjusted R-square* | 0.315 |

Notes: Standard errors are reported in parentheses. *** $p < 0.01$ ** $p < 0.05$ * $p < 0.1$.

## 5. Theoretical and Managerial Implications

### 5.1. Theoretical Implications

This study delivers important theoretical implications in the following three aspects. First, our paper enriches the literature that investigates the effect of consumer ratings. While the extant research has demonstrated its critical role in affecting consumers' value perception [16,17] and their purchase decisions [4,18], little research attention has been paid to how it can be effectively used on the retailer side. Our study aims to address this void by showing that consumer ratings are a critical factor for retailers to determine the optimal discount size.

Second, we derive multiple testable predictions from equilibrium results of our analytical model. Current research reflects that deciding on an appropriate discount size is a challenging task [58]. Our findings demonstrate that consumer ratings affect discount size differently depending on their values and present responding thresholds. Then, we carry out regression analysis using collected data of online audiobooks. To our knowledge, our paper is one of the first attempts to both analytically and empirically check the impact of consumer ratings on retailers' discount decisions.

Third, our study complements prior research on moderating effects [59] by exploring two key moderators, consumers' confidence and the regular price of the digital good. Interestingly, our findings reflect that they have moderating effects in opposite directions. So far, little research effort has been made with regard to these two moderators in the literature. By exploring those moderating effects, our paper enhances the current understanding of retailers' discount promotion strategy.

*5.2. Managerial Implications*

Our study also yields several important practical implications. First, our results show that retailers' discount size is negatively associated with non-supreme consumer ratings of online audiobooks. This finding suggests that retailers need to devote resources towards encouraging consumers to leave favorable ratings, as well as towards keeping these ratings. In return, they can take advantage of these positive ratings by offering a smaller discount size. In addition, retailers can divide their audiobooks into different groups by rating levels and apply the corresponding discount strategy.

Second, we find that there is heterogeneity in the effect of consumers' confidence on the optimal discount size. In particular, the effect is smaller for those audiobooks with higher confidence level (using the number of raters as a proxy). Our results suggest that retailers need to keep improving and maintaining the consumers' confidence. For example, retailers can give monetary and non-monetary rewards to consumers who frequently rate their purchases. In addition, they need to dedicate more resources to quickly respond to and mitigate consumers' criticism. With the development of AI technologies, retailers might consider using big data tools to estimate consumers' confidence level more accurately.

Third, our findings reflect that the effect of consumer ratings on discount size is greater for audiobooks with higher regular prices. Our results demonstrate the importance of taking the regular price into consideration when implementing discount promotion. From a practical standpoint, retailers should offer a smaller discount size for high-price audiobooks with non-supreme consumer ratings. Conversely, for low-price audiobooks with similar ratings, a larger discount is needed.

# 6. Conclusions

Retailers of digital goods often use discount pricing to attract consumers [60]. To maximize their benefit, they naturally wish to integrate useful information when deciding discount size. One key resource of interest is consumer ratings, which play a critical role in affecting consumers' purchase decisions [17,27,28]. Despite growing academic interest, there are surprisingly few studies on how to make use of consumer ratings on the retailer side of the market, which motivates this study.

Our study aims to investigate how retailers can employ online consumer ratings in their decision-making process. We first built a two-period model to analyze their optimal discount size in response to online consumer ratings. Our results show that consumer ratings have a negative impact on the optimal discount size for digital goods with non-supreme ratings. In addition, we found that consumers' confidence has an enlarging moderating role, while the regular price has a reducing moderating role. Then, we conducted an empirical study to test our hypotheses, using the real business data of 419 audiobooks. Our proposed hypotheses are supported by the empirical analysis results.

Compared with previous studies, our paper delivers new insights. For example, Ref. [42] also analyzed how retailers modify their prices in response to consumers' feedback. However, their assumption of different quality does not apply to the digital goods markets. Perhaps that is why the model suggests that retailers offer discounts at the beginning rather than offering discounts at a later stage. Furthermore, prior research has examined different moderators from our study such as exposure to reviews [27], social orientation [31], and review helpfulness [33]. Only ref. [58] has identified a similar moderator of discount credibility. One reason is that we explore the moderating effects from the perspective of retailers' discount decisions, while those studies inspect from the perspective of consumers' purchase intentions. This structural contrast might explain the difference between our findings and theirs.

This paper makes the following contributions. First, while most researchers have only focused on the consumer side, we extend the current studies of consumer ratings to the retailer side. In particular, we examine how retailers determine optimal discount size in response to consumer ratings. Second, our results help retailers understand that consumer ratings effect changes with rating values. Our findings also remind them to consider moderating effects. Third, we present a broad analysis using both an analytical model and an empirical analysis of online audiobooks. So far, relevant empirical research based on real business data is rare in this area. While we focus on audiobooks, our findings can be generalized to other digital goods markets such as online software APP, online newspapers, digital movies, etc.

Our paper has a few limitations. First, we have a relatively small dataset (419 audiobooks), especially for group 2 (audiobooks with supreme ratings). An empirical analysis with a larger sample size would help ensure the reliability of the findings. Second, we make a common assumption of myopic and strategic consumers in our model. Nonetheless, it is challenging to verify that assumption in our collected data. Third, we have not incorporated the subscription programs of audiobooks in our model [61]. Subscribers of audiobooks might have different utilities towards discount promotion. Finally, our data collection is limited because the online rating from Chirpbooks only provides an overall rating. If the platform provided more detailed rating information, such as rating on the content of the book, rating on the narrator, etc., we would be able to perform additional analyses.

Our research can be extended in the following aspects. First, we only focus on the impact of consumer ratings on discount size. Prior research has shown that discount duration can be another critical factor that is worth investigating [62]. Second, we have not collected data of audiobooks without discounts. Retailers do not randomly decide which titles to offer discounts on. Therefore, it is worth investigating what factors will affect retailers' decisions as to whether to offer a discount or not. Third, our dataset does not include consumer ratings with extremely low values. Future research might examine whether the impact of extremely low ratings on discount size is consistent with our predictions. Finally, there are two models between retailers and publishers in the online book market, the wholesale model and the agency model [63]. Researchers could examine the impact of online ratings on retailers' discount strategy under these two different settings. It will be interesting to see what we might observe.

**Funding:** This research received no external funding.

**Institutional Review Board Statement:** Not applicable.

**Informed Consent Statement:** Not applicable.

**Data Availability Statement:** Data available based on requests from the authors.

**Acknowledgments:** The author wants to thank the valuable comments from the 2025 Research seminar of Broadwell College of Business and Economics (BCBE) of Fayetteville State University. They provided valuable input and support for my research.

**Conflicts of Interest:** The author declares no conflicts of interest.

## Appendix A

**Proof of Lemma 1.** When $r \geq \frac{1-(1-\lambda)qe}{\lambda\beta}$, $q_u = q_e + \lambda(\beta r - q_e) \geq 1$. So $q_u = 1$. $U = 1 - (1 - x) \times p - c - ks$, and the demand is $m(1 - (1 - x) \times p - c)/k$. Thus, $\pi_t = \frac{p(q_e - p)}{k} + \frac{m(1-x)\times p \times (1-(1-x)\times p-c)}{k}$. $\frac{\partial \pi_t}{\partial x} = \frac{p(-1+c-2p(-1+x))}{k} = 0$, $x^* = 1 - \frac{1-c}{2p}$.
When $r \leq \frac{(\lambda-1)qe}{\lambda\beta}$, $q_u = q_e + \lambda(\beta r - q_e) \leq 0$. So $q_u = 0$. No demand. $\square$

**Proof of Lemma 2.** In the second time period, $0 < q_u = (q_e + \lambda(\beta r - q_e)) < 1$, $U = (q_e + \lambda(\beta r - q_e))(1 - x) \times p - c - ks$, and the demand is $\frac{m(q_e + \lambda(\beta r - q_e))(1-x) \times p - c}{k}$. $\pi_2 = \frac{m(1-x)p(-c+q_e+\lambda(-q_e+\beta r))-p(1-x))}{k}$. Thus, $\pi_t = \frac{p(-p+q_e)}{k} + \frac{m(1-x)p(-c+q_e+\lambda(-q_e+\beta r))-p(1-x))}{k}$. $\frac{\partial \pi_t}{\partial x} = \frac{mp(c-q_e+\lambda q_e - \lambda \beta r - 2p(-1+x))}{k} = 0$, $x^* = \frac{c+2p-q_e+\lambda q_e - \lambda \beta r}{2p}$. $\square$

**Proof of Proposition 1** (**Effect of consumer ratings on discount size**). Based on Lemma 2 results, we obtain $x^* = \frac{c+2p-q_e+\lambda q_e - \lambda \beta r}{2p}$. Therefore $\frac{\partial x}{\partial r} = -\frac{\lambda \beta}{2p} < 0$. $\square$

**Proof of Proposition 2** (**Moderating effects**). Based on Lemma 2 results, $x^* = \frac{c+2p-q_e+\lambda q_e - \lambda \beta r}{2p}$. Therefore, $\frac{\partial x}{\partial r} = -\frac{\lambda \beta}{2p}$ increases with $p$ and $\frac{\partial x}{\partial r} = -\frac{\lambda \beta}{2p}$ decreases with $\beta$. $\square$

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
