# Peer review of "Incorporating Consumer Ratings in Retailers’ Discount Pricing of Digital Goods"

_jtaer, doi:10.3390/jtaer20040285_

Round 1
Reviewer 1 Report
Comments and Suggestions for Authors
This is very interesting paper in retailing and discounts in digital goods. However, authors can improve the quality of the paper.
Please see comments below:
- Authors need to revise an abstract by writing concisely. The important content should include contribution, concept, methodology, data collection and analysis, results, implications and limitation. Add 1–2 sentences quantifying key effects and stating generalizability/boundary conditions.
- From Research question 1: When and how will consumers' rating be associated with retailers' decision of discount size? Authors set the research question with ‘How’ that make readers expect to see interviews or other qualitative methods. Please redesign the first research question.
- Authors need to mention the contribution at the introduction section by mentioning why retailers, discount size, and consumer rating in this paper can provide the contribution to the field. Additionally, authors need to demonstrate why audiobook was chosen as the context of the study to explore what factors will moderate the effect of consumers' rating on discount size.
- If you claim to be the first to link consumer rating thresholds to retailer discount depth in digital goods, narrow the claim (e.g., “for audiobooks on a non-subscription platform with intermittent price promotions”) and contrast with nearby streams: deal-of-the-day design, digital goods pricing with negligible marginal cost, and algorithmic pricing with reputation signals.
- Authors provided details of analytic model in 3.1 model setup and found 2 types of consumers – myopic and strategic consumers. Were these types of consumers applied to collect data or to develop data collection condition? If yes, how authors can classify the data set for each group?
- Ratings can move with promotions (exposure boosts reviews), and price can influence subsequent ratings. Discuss this and, if feasible, use lags (lag rating/raters relative to the discount day) or an IV proxying exogenous exposure (e.g., exogenous list placements) to reduce simultaneity concerns.
- The dataset contains only already-discounted titles on one retailer. Discount existence is not random; model the probability of being discounted first (e.g., Probit/Logit selection or control function), then discount depth conditional on selection. This addresses the fact that your DV is observed only when retailers choose to discount. (You state the platform choice and period; see data description.)
- Authors need to provide contributions clearly in the discussion section.

Author Response
Please see the attached response.

Reviewer 2 Report
Comments and Suggestions for Authors
Although the authors have provided an innovative and interesting research viewpoint, the following issues are not exactly expressed, however, thus calling for further explanation.
- Consistency and rigidness of the research framing process
- According to the draft, it is hard to see the tight relationship between Stage 1 and Stage 2 (i.e., even if Stage 1 is neglected, the study can still work well based on Stage 2). It, therefore, calls for further explanation in the necessity, importance and contribution of executing Stage 1.
- There exists inconsistent casualty amongst the proposed Lemmas, the conceptual model, and the hypotheses (i.e., the conceptual moderating effects drawn on Fig. 2 are different from that of H2 and H3), thus calling for revision.
- How the research process of Stage 2 can efficiently represent the framework drawn on Fig. 1 is also questioned.
- Finally, although the model settings (including the total number of samples) are believed supported by cited references, the authors have to further explain whether the result drawn based on the extreme small sample size of group 2 (i.e., N=31) truly fulfill the requirements of model reliability and validity (i.e., showing that the cited studies also ensuring research reliability by adopting similar approaches for sample division and testing).
- Generalizability of the proposed model
- The authors provide limited information in terms of why audiobooks are appropriate as typical digital goods for analysis; as well, how the findings can be applied to digital goods (except for audiobooks) are not examined and explained. Thus, the authors should provide further explanation in this regard.
- Minor issues for revision
- To ensure and to spotlight the applied context, the authors are suggested to verify whether the research target is online retailer or retailer (owing to the fact that the multi-channel setting and pricing of a typical retailer may affect the robustness of the proposed assumption).
- Appropriateness of the proposed assumption: Although the authors provide references for shaping assumption 3, arguments that the casual pattern of consumers purchasing digital goods as gifts for friends may damage this key assumption, thus calling for further explanation.
- With regard to the hypotheses, in order to ensure the generalizability of the proposed model, the authors are suggested to revise the wording (i.e., the term audiobooks should not be used directly in the general hypotheses).
Author Response
Please see the attached response.

Reviewer 3 Report
Comments and Suggestions for Authors
I was pleased to read the article:
“Incorporating Consumers’ Rating in Retailers’ Discount Pricing 2 of Digital Goods”.
It covers an important research issue that is rarely addressed by other researchers.
In my opinion, the following elements are worth considering and modifying:
The abstract (lines 8-23) should clearly state the purpose of the article.
The Introduction should provide examples of sales, prices and discounts for selected digital goods sold/purchased online.
The purpose of the article should be added at the end of the Introduction (lines 269-279).
The justification given in Chapter 4 for the selection of audiobooks for the study is too superficial. It is not convincing (lines 269-279). The place of audiobooks among digital goods sold/purchased online according to the criteria adopted by the authors should be presented in more detail.
The Conclusions section contains repetitions (lines 473-490). It is worth considering whether they are necessary here. Instead, the focus should be on the conclusions of the study. This section also includes a brief Discussion. It is worth expanding on this, even though the authors note that similar studies are very rare. The discussion should refer to the results of other authors' studies and indicate whether the results presented in the article are similar or different. The discussion should also include the reasons for the differences and similarities between the compared results.

Author Response
Please see the attached response file.

Round 2
Reviewer 1 Report
Comments and Suggestions for Authors
Thank you very much for your careful revision and for addressing all the concerns raised during the previous review round. I have thoroughly evaluated the revised manuscript and find that the authors have satisfactorily responded to all comments.
I am pleased to recommend acceptance of the manuscript for publication. I appreciate the authors’ effort and diligence in improving the manuscript.
Reviewer 2 Report
Comments and Suggestions for Authors
- There still lacks direct linkage amongst Figure 1, Figure 2, variable settings. and the research outcome / hypotheses. More specifically,
- How the authors form Figure 2 based upon Stage 2 of Figure1 is unknown (including using inconsistent wordings for the outcome)
- How the variables shown in Table 3 represent for constructs shown in Figure 2 is known (i.e., which variables representing constructs “consumers’ confidence” and “regular price” are not drawn)
- According to Table 6, the authors have to explain why the 2 moderating constructs shown in Figure 2 are independent (i.e., not inter-related) and the consistency of model testing
- Details of the casual model lacks theoretical support and convinced validation process, thus calling for further examination
- How the variables shown in Table 3 are selected, as well as why they are suitable for forming constructs shown in Figure 2 are unknown (i.e., lacking theoretical support in the Literature review section)
- How to validate the hypotheses are also unexplained in detail (e.g., if Ln_Length and Ln_BookAge are variables representing the construct “consumers’ confidence,” the authors have to explain how to validate and examine the casualty relationships exist)
- As well, appropriateness of why the research adopts current practices for data transforming (i.e., converting into log format) is not explained
- Although the context and limitation of sample size have been well documented, it is still argued that the appropriateness of applying the extreme small sample size of group 2. The authors are required to provide hard evidence in telling that such issue is not a key challenge for proving hypotheses in the past research.
- Minor issue: Be sure what have been discussed in Literature section (including the review shown in Table 1) and what have been formulated in the Analytical Model section (i.e., key notations and variables in the mathematical models and Table 2 truly relevant to the examined model shown in Figure 2)

Round 3
Reviewer 2 Report
Comments and Suggestions for Authors
- With regard to Section 4.1
- The authors are required to provide more solid / hard evidence / support in explaining why the 4 proposed control variables are applied or used (i.e., age of a book [Ln_Bookage], length of audiobook [Ln_Length], and 2 dummy variables).
- Besides, be sure the proposal casual model is consistent with the analytical model being applied. More specifically, regarding that H2 and H3 represent moderating effects of the model, the authors are required to explain why Model 1 is appropriate (since the 3 variables are reviewed as independent variables. In contrast, if Model 1 represents the truly casual model from the authors’ perspective, the research model (i.e., Figure 2) and the wordings / statements of H2 and H3 are required to redraw.
- With regard to Table 5 and Table 6, the authors are suggested to provide further explanation in terms of the explanatory power / effects of key variables relevant to hypotheses (i.e., Rating, Ln_Raters, and Ln_Regprice) and that of control variables (i.e., coefficients of the 4 control variables)
- Moreover, the appropriateness of data distribution of the first 5 variables shown in Table 3 is suggested to be explained.
- Finally, to ensure the statement consistency, the sequence Ln_Bookage and Ln_Length is suggested to exchange (i.e., in the paragraph, Ln_Bookage shows earlier than Ln_Length, whereas in Table 3 the showup sequence is in a reverse manner)
- Be sure the appropriateness of adding the clarification “Please note that Figure 2 is based on our analytical results, not on Figure 1” in the “Analytical Model” section.
